# The usefulness of change in CT score for evaluating the activity of *Mycobacterium abscessus* (Mab) pulmonary disease (Mab-PD)

**Kyota Shinfuku[1]◉, Hiromichi Hara📧[2]◉\*, Naoki Takasaka[1], Takeo Ishikawa[1], Jun Araya[2], Kazuyoshi Kuwano[2]**

**1** Division of Respiratory Diseases, Department of Internal Medicine, The Jikei University Daisan Hospital, Tokyo, Japan, **2** Division of Respiratory Diseases, Department of Internal Medicine, The Jikei University School of Medicine, Tokyo, Japan

◉ These authors contributed equally to this work.
\* hirohara@jikei.ac.jp

## Abstract

### Objectives

Semi-quantitative CT score is generally used for evaluating the disease status of *Mycobacterium abscessus (Mab)* Pulmonary disease (Mab-PD). However, its accuracy and clinical usefulness are limited, since the CT score is largely affected by coexisting lung disease. Hence, we hypothesized that numerical change in CT score during the observation period may be useful for evaluating disease activity of Mab-PD.

### Methods

Patients diagnosed with Mab-PD based on the official ATS/ERS/ESCMID/IDSA statement at Jikei University Hospital and Jikei Daisan Hospital between 2015 January 1 and 2021 July 31 were included (n = 32). We reviewed the medical records, and bacteriological and laboratory data of the patients. Chest CT was performed at diagnosis in all 32 cases. In 18 cases, chest CT images within 4 years before diagnosis were available. The numerical change in CT score between two time points was calculated and the association of the CT scores with sputum Gaffky score and serum CRP was examined.

### Results

CT score at diagnosis was not correlated with sputum Gaffky score nor serum CRP, while the difference of absolute value and change rate in CT score between at diagnosis and immediate past CT were well correlated with both sputum Gaffky score and serum CRP.

### Conclusions

Chronological change in CT score may more precisely reflect the disease activity of airway mycobacterial burden and systemic inflammation in Mab-PD at the timing of diagnosis.

**Data Availability Statement:** Data cannot be shared publicly because of privacy of the patients. Data from this study are available upon request (Hiromichi Hara, E-mail: hirohara@jikei.ac.jp or

ethics committee of Jikei University, E-mail:
crb@jikei.ac.jp).

**Funding:** he author(s) received no specific funding
for this work.

**Competing interests:** The authors have declared
that no competing interests exist.

**Abbreviations:** Mab, Mycobacterium abscessus;
MAC, M.avium complex; NTM, non-tuberculous
mycobacteria; PD, Pulmonary disease.

## Introduction

In recent years, the incidence of non-tuberculous mycobacterial (NTM) infections increases
worldwide, including in Japan [1, 2]. Among them, *Mycobacterium abscessus* (Mab) infections
are rapidly increasing. Mab is one of the rapidly growing NTM groups [3, 4]. It belongs to s
group IV in Runyon classification, composed of three subspecies (*M. abscessus subsp. absces-
sus*, *M. abscessus subsp. massiliense and M. abscessus subsp. bolletii.*) [3]. Mab is ubiquitous in
the environment including soil and tap water, which causes chronic infection in skin, bone,
and respiratory organs in humans [5]. Mab Pulmonary disease (Mab-PD) is generally treated
with combinations of several intravenous and oral agents based on its disease activity [3, 6, 7].

Although an Official ATS/ERS/ESCMID/IDSA Clinical Practice Guideline for NTM pulmo-
nary disease was published in 2020 [3], a variety of clinical issues including evaluation of disease
activity, selection of medical agents, timing of treatment, and duration of treatment remain
unresolved. Evaluation of disease activity of Mab-PD is the initial and important step for decid-
ing the administration of treatment. The activity of Mab-PD is commonly evaluated by the
patient's symptoms, serum CRP, amount of mycobacteria in sputum, and CT findings. How-
ever, symptoms are not quantitative; CRP is less sensitive. amount of mycobacteria in sputum is
highly variable, in addition, sputum samples cannot be obtained timely. Hence, semi-quantita-
tive CT score evaluating severity and extent of bronchitis, bronchiectasis, cavity, nodule, and
consolidation is generally used for assessing disease activity and severity for Mab-PD [8]. How-
ever, its clinical usefulness is limited, since these CT findings are not specific for Mab-PD, and
the score can be affected by coexisting lung disease [9–11]. Accordingly, in comparison to a sin-
gle evaluation of CT score at the timing of diagnosis, we hypothesized that numerical change in
CT score during the observation period may be useful for precisely evaluating disease activity of
Mab-PD. In this study, we examined the clinical usefulness of CT score at diagnosis and its
chronological change from prior to diagnosis to the timing of diagnosis of Mab-PD.

## Methods

### Study subjects

This study is a retrospective cohort study of patients diagnosed with Mab-PD based on the
official ATS/ERS/ESCMID/IDSA statement [3] at Jikei University Hospital and Jikei Daisan
Hospital from January 1, 2015 (01/01/2015) to July 31, 2021 (07/31/2021). Patients were eligi-
ble for the study if they met the following inclusion criteria 1) Age $\geq$ 20 years old, 2) patients
diagnosed with Mab-PD based on the official ATS/ERS/ESCMID/IDSA statement.

A total of 32 patients were included in this study. Among them, 13 patients were included
in our previous report [12]. This study was approved by the Ethics Committee of Jikei Univer-
sity (33–219 (10836)), and adhered to the tenet of medical ethics. We performed opt-out con-
sent on the website of our hospital, based on the ethical guidelines of Jikei University.
(Informed consent was not necessary for this retrospective study). Patients' medical records
from 2015 (01/01/2015) to July 31, 2021 (07/31/2021) were accessed. All data were fully anon-
ymized after collecting from the medical records and used for further statistical analysis. Chest
CT images were evaluated by two specialists of the Japanese Respiratory Society (H.H. and K.
S.). Data from this study are available upon request (Hiromichi Hara, E-mail: hirohara@jikei.
ac.jp or ethics committee of Jikei University, E-mail: crb@jikei.ac.jp).

### Diagnosis of Mab-PD

Respiratory samples from patients with clinical suspicion of NTM pulmonary disease were cul-
tured on Ogawa media or Mycobacteria Growth Indicator Tube, and species of cultured

mycobacteria were determined by mass spectrometry or DDH (SRL, Inc. laboratory, Tokyo, Japan). According to the official ATS/ERS/ESCMID/IDSA statement, a diagnosis of Mab-PD was made [3].

## Data collection

We retrospectively reviewed the medical records of the enrolled patients. Clinical characteristics including age, sex, BMI, smoking history, comorbidities, coexisting lung disease, and Gaffky score in sputum were investigated. The maximum number of Gaffky scores (Gaffky score; a scale assessing the number of mycobacteria bacilli present, 0–10) at diagnosis was selected [13]. CT score was calculated using a scoring system for Mab-PD [8] by HH or KS. A total score of 30 was allocated for five types of lung lesions including bronchiectasis (Severity;3 points, Extent;3 points, Mucus plugging; 3 points, total 9 points), bronchiolitis (Severity;3 points, Extent;3 points, total 6 points), cavity (Diameter; 3 points, Wall thickness; 3 points, Extent;3 points, total 9 points), nodule (3 points), and consolidation (3 points) depending on the severity of lung lesions [8]. The difference of absolute value and change rate in CT score (based on the difference of CT scores between at diagnosis and immediate past CT score) were calculated.

## Statistical analysis

Simple linear regression analysis was performed to estimate the association of CT score at diagnosis of Mab-PD with sputum Gaffky score, and serum CRP. The association of changes in two time points CT scores with sputum Gaffky score, serum CRP was also examined. All statistical analyses were performed using GraphPad Prism version 8.4.3 for Macintosh (Graph-Pad Software La Jolla, CA, USA), and the statistical significance was set at $P < 0.05$.

## Results

### Clinical background of Mab-PD (Table 1)

The Clinical backgrounds of all Mab-PD patients were shown (n = 32, Table 1). Twenty-eight patients (88%) had coexisting pulmonary diseases such as bronchiectasis, prior pulmonary tuberculosis, obstructive pulmonary disease (chronic obstructive pulmonary disease, Asthma, bronchiolitis obliterans) and interstitial lung disease. Serology showed no or slight elevation of white blood cell counts and CRP levels.

### CT score at diagnosis of Mab-PD lung lesions (Table 2)

The severity and extent of 'bronchiectasis' and 'cellular bronchiolitis' were higher than the other parameters. 'Bronchiectasis' and 'cellular bronchiolitis' were characteristic findings of Mab-PD. The mean total score was 9.8 (Max 30).

### Correlation of CT score at the timing of diagnosis with sputum Gaffky score and serum CRP

Correlation of CT score at the timing of diagnosis with sputum Gaffky score (Fig 1A) and CRP (Fig 1B) are shown. No significant correlations were demonstrated in either case.

### Correlation of chronological change in CT score with sputum Gaffky score and serum CRP

In 18 patients of 32 patients, chest CT was performed for follow-up of other diseases within 4 years before the diagnosis of Mab-PD (Table 2). The numerical changes in CT score between

**Table 1. Characteristics of patients and CT scores at diagnosis.**

| | Mab-PD (n = 32) |
|---|---|
| Age (years)[a] | 67.5 ± 14.3 |
| Male[b] | 13 (41) |
| BMI[a] | 19.1 ± 3.5 |
| History of smoking[b] | 9 (28) |
| Underlying pulmonary disease[b] | 28 (88) |
| Bronchiectasis | 14 (34) |
| Prior pulmonary tuberculosis | 1 (3) |
| Obstructive pulmonary disease | 6 (19) |
| Chronic obstructive pulmonary disease | 3 (9) |
| Bronchial asthma | 2 (6) |
| Bronchiolitis obliterans | 1 (3) |
| Interstitial lung disease | 7 (22) |
| Malignancy | 8 (25) |
| Autoimmune disease | 2 (6) |
| Use of steroid or immunosupressant[b] | 9 (28) |
| Use of inhaled corticosteroid[b] | 4 (13) |
| Number of times of AFB culture positive[c] | 3 (0–9) |
| Maximum Gaffky scale[c] | 2 (0–10) |
| Gafky scale ≧ 5[b] | 6 (19) |
| positive culture of other NTM in sputum at different time[b] | 12 (38) |
| positive culture of *Aaspergillus* in sputum[b] | 6 (19) |
| Laboratory data at diagnosis [d, c] | |
| White blood cell count (3300–8600) (/μL) | 7800 (2800–22300) |
| Neutrophil count (1700–6300) (/μL) | 5350 (1500–21500) |
| lymphocyte count (1000–3100) (/μL) | 1600 (40–3800) |
| CRP (0.14 >) (mg/dl) | 0.3 (0–9.6) |
| Alb (4.1–5.1) (g/dl) | 3.8 (1.7–4.5) |
| CT score at diagnosis [a] | |
| Bronchiectasis | |
| Severity | 1.6 ± 0.7 |
| Extent | 1.6 ± 0.8 |
| Mucus plugging | 0.5 ± 0.5 |
| Cellular bronchiolitis | |
| Severity | 1.6 ± 1.1 |
| Extent | 1.2 ± 0.8 |
| Cavity | |
| Diameter (cm) | 0.5 ± 0.8 |
| Wall thickness(mm) | 0.6 ± 1.0 |
| Extent | 0.4 ± 0.7 |
| Nodules | 0.5 ± 0.5 |
| Consolidation | 1.1 ± 0.7 |
| total scores | 9.8 ± 3.2 |

Date are presented as

[a] mean ± SD

[b] n (%), or

[c] median (minimum-maximum).

[d] normal range.

BMI: body mass index, AFB: acid-fast bacilli, Mab: *Mycobacterium abscessus*

NTM: nontuberculous mycobacteria, CRP: c-reactive protein

**Table 2. Change of CT scores.**

| Mab-PD (n = 18) | Data at diagnosis | Data at pre-CT scan | Change of CT scores |
|---|---|---|---|
| Age (years)[a] | 63.9 ± 12.6 | ••• | ••• |
| Male[b] | 7 (39) | ••• | ••• |
| Underlying pulmonary disease[b] | 18 (100) | ••• | ••• |
| Bronchiectasis | 9 (50) | ••• | ••• |
| Obstractive pulmonary disease | 5 (28) | | |
| Chronic obstructive pulmonary disease | 2 (11) | ••• | ••• |
| Bronchial asthma | 2 (11) | ••• | ••• |
| Bronchiolitis obliterans | 1 (6) | ••• | ••• |
| Interstitial lung disease | 4 (22) | ••• | ••• |
| Time from pre-CT to diagnosis CT (days) | 697 ± 381 | ••• | ••• |
| CT scores [a] | | | |
| Bronchiectasis | | | |
| Severity | 1.6 ± 0.6 | 1.4 ± 0.7 | 0.1 ± 0.3 |
| Extent | 1.6 ± 0.7 | 1.3 ± 0.7 | 0.2 ± 0.5 |
| Mucus plugging | 0.6 ± 0.6 | 0.4 ± 0.5 | 0.1 ± 0.3 |
| Cellular bronchiolitis | | | |
| Severity | 1.7 ± 1.0 | 1.0 ± 0.8 | 0.6 ± 1.0 |
| Extent | 1.2 ± 0.8 | 0.8 ± 0.6 | 0.3 ± 0.5 |
| Cavity | | | |
| Diameter (cm) | 0.5 ± 0.9 | 0.2 ± 0.5 | 0.2 ± 0.7 |
| Wall thickness (mm) | 0.6 ± 1.0 | 0.3 ± 0.7 | 0.2 ± 0.7 |
| Extent | 0.4 ± 0.7 | 0.2 ± 0.5 | 0.1 ± 0.5 |
| Nodules | 0.3 ± 0.4 | 0.3 ± 0.4 | 0.0 ± 0.3 |
| Consolidation | 1.1 ± 0.6 | 0.6 ± 0.5 | 0.3 ± 0.5 |
| total | 9.8±2.8 | 7.0±3.3 | 2.7 ± 2.9 |

Date are presented as [a]mean ± SD, [b] n (%) and cmedian (minimum-maximum)

Mab: *Mycobacterium abscessus*, PD: pulmonary disease, CRP: c-reactive protein.

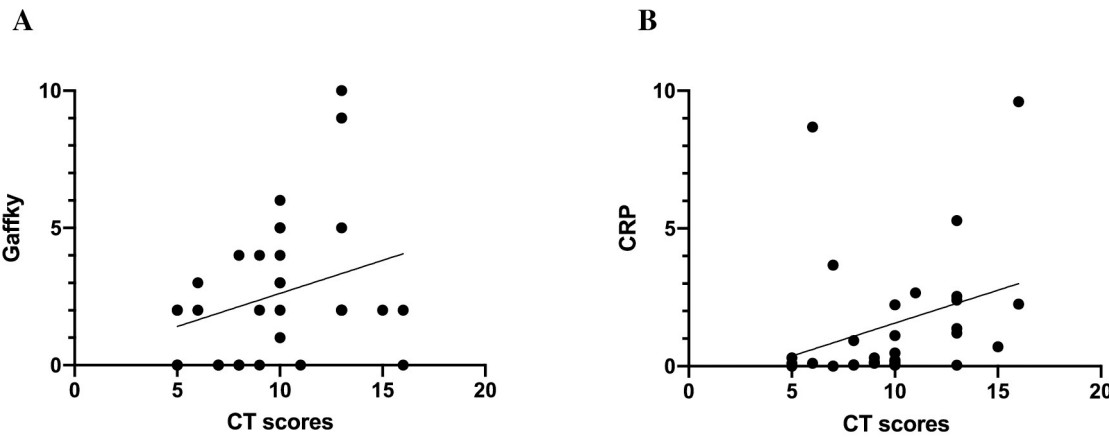

Y= 0.2407 × X + 0.208 : R squared =0.0923 (*p*= 0.09)

Y= 0.239 × X- 0.8257 : R squared =0.1035 (*p*= 0.076)

**Fig 1. Relationship between CT score at diagnosis and sputum Gaffky score, serum CRP.** Correlation between CT score at the timing of diagnosis with sputum Gaffky score (Fig 1A) and CRP (Fig 1B) are shown.

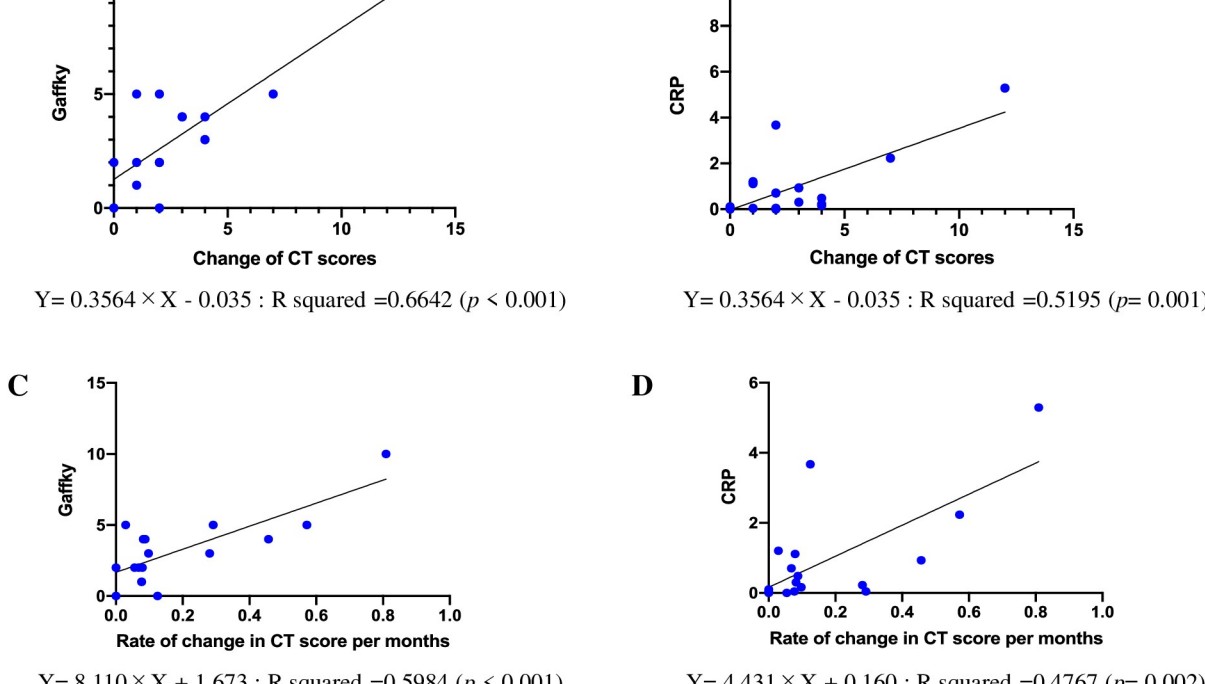

Y= 0.3564 × X - 0.035 : R squared =0.6642 (*p* < 0.001)

Y= 0.3564 × X - 0.035 : R squared =0.5195 (*p*= 0.001)

Y= 8.110 × X + 1.673 : R squared =0.5984 (*p* < 0.001)

Y= 4.431 × X + 0.160 : R squared =0.4767 (*p*= 0.002)

**Fig 2. Relationship between chronological change in CT score and sputum Gaffky score, serum CRP.** Correlation between the difference of CT scores with sputum Gaffky score (Fig 2A) and CRP (Fig 2B) are shown. The change rate in CT score was also correlated with sputum Gaffky score (Fig 2C) and CRP (Fig 2D), respectively.

at diagnosis and immediate past CT score was calculated. The difference in CT score between two time points was well correlated with sputum Gaffky score (Fig 2A) and CRP (Fig 2B), respectively. The largest difference was observed in the category of severity in 'cellular bronchiolitis' (previous CT:1.0, CT at diagnosis: 1.7). Next, the change rate in CT score was calculated by dividing by month, which was also correlated with sputum Gaffky score (Fig 2C) and CRP (Fig 2D), respectively.

## Discussion

In this study, we demonstrated that CT score at diagnosis was not correlated with sputum Gaffky score and serum CRP, but the difference of absolute value and change rate in CT score between at diagnosis and immediate past CT was well correlated with both sputum Gaffky score and serum CRP. Accordingly, chronological change in CT score may more precisely reflect the disease activity of airway mycobacterial burden and systemic inflammation in Mab-PD.

Mab colonizes airway or lung tissue, especially in the setting of disease complicated lungs after inhalation of the pathogen. Decreased local or systemic host defense may affect the subsequent distribution of the mycobacteria into lung tissues, resulting in Mab-PD development. When Mab is detected in the respiratory specimens, it is critically important to determine whether simple colonization or Mab-PD development for treatment administration at the appropriate timing. Although a semi-quantitative CT score is generally used for assessing Mab-PD activity [9], its clinical usefulness is limited, since the score is affected by high frequent coexisting lung disease. Indeed, in the present study, CT score at diagnosis did not correlate with sputum Gaffky score or serum CRP. However, the difference and change rate of CT score between at diagnosis and immediate past CT score (chronological change in CT score),

significantly correlated with both the sputum Gaffky score and CRP. It has been recognized that sputum Gaffky score reflects bacteriological activity and CRP reflects the serological activity. It is plausible that chronological change in CT score can be an important indicator reflecting both bacteriological and serological activity of Mab-PD. Accordingly, it is important to evaluate the CT score at multiple time points at the timing of diagnosis and chronological change should be used as an indicator of radiological activity if previous CT scans were available. Chronological change of CT scores might be useful in assessing disease activity in other NTM-PDs, which should be elucidated in future studies. Progression of PD might at least partly depend on growth speed of the mycobacteria, hence, we speculate that change of CT scores of rapidly growing mycobacteria including Mab might be more detectable than slowly growing mycobacteria including MAC.

Among the parameters of CT evaluation, the largest difference in severity was observed in the category of 'cellular bronchiolitis'. Since 'cellular bronchiolitis' is not necessarily accompanied by tissue destruction, CT findings might be promptly changeable reflecting disease activity, compared to other parameters with severe tissue destruction including 'bronchiectasis' and 'cavitation'.

There are several limitations of this study. First, this retrospective study included a small number of cases, which can be attributed to the relatively small number of Mab-PD cases compared to MAC-PD cases in Japan. And, the results in this study could be affected by an outlier effect. Data from a single patient were separated from the rest of the patients. We reviewed in detail the medical record of the patient, and concluded that delayed diagnosis led to advanced pulmonary involvement and elevated sputum bacterial counts and serum CRP. However, no obvious reason was found to exclude the patient from this study. When outlier tests were performed by InterQuartile Range (IQR) method(Tail Quantile 0.1 Q3) and Robust estimation method (Huber M-estimation K sigma 4) using JMPPro16 Software (SAS Inst. Inc., Cary, NC), no outliers were found. Hence, we included this patient in the study. (Even after excluding that patient, the change in CT scores was still significantly correlated with the sputum Gaffky score. However, there was no significant difference in the association between the change in CT scores and CRP if the patient was excluded. CRP might be less likely to rise in patients with mild lesions, and was less likely to reflect disease activity than sputum Gaffky score.) Second, previous CT scans were not available in around half of the cases in this study and CT score at the timing of diagnosis did not reflect the disease activity of Mab-PD. Third, due to the retrospective nature of this study, the interval between CT scans was variable in each case, which may have an impact on not only the difference in CT score but also on the usefulness for predicting disease activity. Although our results suggest that difference in CT score in the setting of less than a 4-year interval may reflect disease activity, we speculate that the change rate in CT score under short-term observation will be a more accurate indicator of Mab-PD activity, which should be examined in a future validation cohort study. Progression of other lung diseases than Mab-PD might affect chronological change of CT scores, which was not clinically evident during this observation period.

## Conclusions

Chronological change in CT score may more precisely reflect disease severity of airway mycobacterial burden and systemic inflammation in Mab-PD at the timing of diagnosis.

## Supporting information

**S1 Fig. Samples of CT scoring (Bronchiectasis, bronchiolitis, cavity).**
(TIF)

## Author Contributions

**Conceptualization:** Kyota Shinfuku, Hiromichi Hara, Naoki Takasaka, Takeo Ishikawa, Jun Araya.

**Data curation:** Kyota Shinfuku, Hiromichi Hara.

**Formal analysis:** Hiromichi Hara.

**Methodology:** Hiromichi Hara.

**Project administration:** Kyota Shinfuku.

**Resources:** Kyota Shinfuku, Hiromichi Hara, Naoki Takasaka.

**Software:** Kyota Shinfuku, Hiromichi Hara.

**Supervision:** Kyota Shinfuku, Hiromichi Hara, Naoki Takasaka, Takeo Ishikawa, Jun Araya, Kazuyoshi Kuwano.

**Validation:** Kyota Shinfuku, Hiromichi Hara, Naoki Takasaka, Takeo Ishikawa, Jun Araya, Kazuyoshi Kuwano.

**Visualization:** Kyota Shinfuku, Hiromichi Hara, Jun Araya, Kazuyoshi Kuwano.

**Writing – original draft:** Kyota Shinfuku, Hiromichi Hara.

**Writing – review & editing:** Hiromichi Hara, Jun Araya.

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
