## [Decision Letter · Decision Letter 0]

19 Dec 2022

PONE-D-22-29888The usefulness of change in CT score for evaluating the activity of Mycobacterium abscessus (Mab) pulmonary disease (Mab-PD)PLOS ONE

Dear Dr. Hara,

Thank you for submitting your manuscript to PLOS ONE. After careful consideration, we feel that it has merit but does not fully meet PLOS ONE’s publication criteria as it currently stands. Therefore, we invite you to submit a revised version of the manuscript that addresses the points raised during the review process. Please submit your revised manuscript by Feb 2, 2023. If you will need significantly more time to complete your revisions, please reply to this message or contact the journal office at plosone@plos.org. Please include the following items when submitting your revised manuscript:A rebuttal letter that responds to each point raised by the academic editor and reviewer(s). You should upload this letter as a separate file labeled 'Response to Reviewers'.A marked-up copy of your manuscript that highlights changes made to the original version. You should upload this as a separate file labeled 'Revised Manuscript with Track Changes'.An unmarked version of your revised paper without tracked changes. You should upload this as a separate file labeled 'Manuscript'.

We look forward to receiving your revised manuscript.

Kind regards,

Frederick Quinn

Academic Editor

PLOS ONE

Journal Requirements:

Reviewers' comments:

Reviewer's Responses to Questions

**Comments to the Author**

1. Is the manuscript technically sound, and do the data support the conclusions?

Reviewer #1: Partly

Reviewer #2: Partly

2. Has the statistical analysis been performed appropriately and rigorously? 

Reviewer #1: No

Reviewer #2: I Don't Know

3. Have the authors made all data underlying the findings in their manuscript fully available?

Reviewer #1: No

Reviewer #2: No

4. Is the manuscript presented in an intelligible fashion and written in standard English?

Reviewer #1: Yes

Reviewer #2: Yes

5. Review Comments to the Author

Reviewer #1: Reviewer’s response

The manuscript written by Hara H et al investigates the usefulness of CT score change in characterizing M. abscessus-associated pulmonary disease in human patients. M. abscessus lung infections represent a significant medical burden and a better measurement of Mabs-PD is needed to improve clinical diagnosis and treatment decisions. Lung CT scores alone remain semi-quantitative. The investigators of this short report analyzed retrospective data obtained on a limited patient cohort (n=32). CT changes could be measured over a 4-year period of time in case of 18 patients and found significant correlations between CT score changes and microbiological or immunological measures of the disease. While the manuscript raises an important question and explores an interesting idea, the presented data are not very convincing as they could be affected by an outlier effect, especially if derived from a single patient. This raises serious concerns and a more detailed analysis of the presented data is needed for a better-funded conclusion about this interesting topic.

Please find my minor and major comments below.

Major comments

- While the CT score changes significantly correlate with CRP and Gaffky scores (figure 2), the data are only derived from a handful of patients and could be largely driven by a single outlier. Each panel in figure 2 has a data point in the upper-right corner that stands alone and separated from the rest of the patients. Do these data point belong to the same single patient or multiple individuals? If these data originate from the same patient, they could easily bias the presented data and question whether there is a real correlation between the CT score change (per month or over the entire period of time) and the microbiological/immunological measures.

- Would the data remain significantly correlated in each of the four panels in figure 2 if the upper/right outliers are omitted from the analysis?

Minor comments

- Fonts in table 2 are very small and should be increased for better visibility

- Which statistical method was used to establish the significance of the regression analysis?

Reviewer #2: General:

This is a clinical paper suggesting that scoring of two chest computed tomography (CT) scans done months before and then at the time of diagnosis of M. abscessus pulmonary disease, particularly for measures of bronchiectasis and bronchiolitis, can aide in the diagnosis. This is a retrospective review and the authors do a good job of highlighting the limitation of this study. Since the chest CT scans done prior to the diagnosis were performed for clinical reasons other than assessing for the presence of M. abscessus pulmonary disease and in only a little more than half of the infected patients, it is hard to see how this tool can be used clinically to detect M. abscessus earlier. One piece of data that might be helpful is to assess repeat pulmonary function testing in these patients, which might be more readily available than repeat chest CT scans, in order to see whether decline in lung function is correlated with increase in CT score.

Specific:

Result section Lines 151 to 159. There is no need to repeat all the diagnoses in Table 1. Rather just say that 88% of the cohort had underlying pulmonary disease and leave it to the reader to see the distribution of disease in the table. Also for the lab values, it would be important to include the normal range.

Gaffky score. Please provide a reference for the Gaffky score. Also, in Table 1, I think the line saying “Maximum number of Gaffky scale” should read the “Number of patients that had the maximum Gaffky scale of 10” and it is unclear what “total AFB culture positive count for Mab” means.

Table 2. The upper part of Table 2 repeats much of what is in Table 1 and needs to be deleted. In other words, begin Table 2 with “Time from pre-CT to diagnosis CT (days)” and then have the comparison between the two CT scan scores.

6. PLOS authors have the option to publish the peer review history of their article (what does this mean?). If published, this will include your full peer review and any attached files.

Reviewer #1: No

Reviewer #2: No

---

## [Author Response · Author response to Decision Letter 0]

27 Dec 2022

Reviewer #1

Comment 

The manuscript written by Hara H et al investigates the usefulness of CT score change in characterizing M. abscessus-associated pulmonary disease in human patients. M. abscessus lung infections represent a significant medical burden and a better measurement of Mabs-PD is needed to improve clinical diagnosis and treatment decisions. Lung CT scores alone remain semi-quantitative. The investigators of this short report analyzed retrospective data obtained on a limited patient cohort (n=32). CT changes could be measured over a 4-year period of time in case of 18 patients and found significant correlations between CT score changes and microbiological or immunological measures of the disease. While the manuscript raises an important question and explores an interesting idea, the presented data are not very convincing as they could be affected by an outlier effect, especially if derived from a single patient. This raises serious concerns and a more detailed analysis of the presented data is needed for a better-funded conclusion about this interesting topic. Please find my minor and major comments below.

Major comments

- While the CT score changes significantly correlate with CRP and Gaffky scores (figure 2), the data are only derived from a handful of patients and could be largely driven by a single outlier. Each panel in figure 2 has a data point in the upper-right corner that stands alone and separated from the rest of the patients. Do these data point belong to the same single patient or multiple individuals? If these data originate from the same patient, they could easily bias the presented data and question whether there is a real correlation between the CT score change (per month or over the entire period of time) and the microbiological/immunological measures.

- Would the data remain significantly correlated in each of the four panels in figure 2 if the upper/right outliers are omitted from the analysis?

Response

Thank you for thoughtful comments. The data separated from the rest of the patients belong to the same single patient. We reviewed in detail the medical record of the patient, and concluded that delayed diagnosis led to advanced pulmonary involvement and elevated sputum bacterial counts and serum CRP, however, no obvious reason was found to exclude the patient from this study. When outlier tests were performed by Inter　Quartile Range (IQR) method　(Tail Quantile 0.1 Q3) and Robust estimation method (Huber M-estimation K sigma 4) using JMP　Pro16 Software (SAS Inst. Inc., Cary, NC), no outliers were found. Based on the above, we would like to include this patient in the　study. However, the reviewer's point is very important and fundamental, hence, we excluded the patient who appeared to be an outlier and performed the statistical analysis again according to the reviewer’s comment. Even after excluding that patient, the change in CT scores was still significantly correlated with the sputum bacterial count. However, there was no significant difference in the association between the change in CT scores and CRP if the patient was excluded. The reason for this may be that CRP is less likely to rise in patients with mild lesions. We added the following sentences to the Discussion section (Line 230-242).

And, the results in this study could be affected by an outlier effect. Data from a single patient were separated from the rest of the patients. We reviewed in detail the medical record of the patient, and concluded that delayed diagnosis led to advanced pulmonary involvement and elevated sputum bacterial counts and serum CRP. However, no obvious reason was found to exclude the patient from this study. When outlier tests were performed by Inter　Quartile Range (IQR) method(Tail Quantile 0.1 Q3) and Robust estimation method (Huber M-estimation K sigma 4) using JMP　Pro16 Software (SAS Inst. Inc., Cary, NC), no outliers were found. Hence, we included this patient in the study. (Even after excluding that patient, the change in CT scores was still significantly correlated with the sputum Gaffky score. However, there was no significant difference in the association between the change in CT scores and CRP if the patient was excluded. CRP might be less likely to rise in patients with mild lesions, and was less likely to reflect disease activity than sputum Gaffky score.)

Minor comments

- Fonts in table 2 are very small and should be increased for better visibility

Response

We apologize for small fonts in table2. We increased the font size from 12 to 16.

Comment 

- Which statistical method was used to establish the significance of the regression analysis?

Response

We added the following sentences in the Methods section. (Line141-142) 

Simple linear regression analysis was performed to estimate the association of CT score at diagnosis of Mab-PD with sputum Gaffky score, and serum CRP.

Reviewer #2: General:

Comment 

This is a clinical paper suggesting that scoring of two chest computed tomography (CT) scans done months before and then at the time of diagnosis of M. abscessus pulmonary disease, particularly for measures of bronchiectasis and bronchiolitis, can aide in the diagnosis. This is a retrospective review and the authors do a good job of highlighting the limitation of this study. Since the chest CT scans done prior to the diagnosis were performed for clinical reasons other than assessing for the presence of M. abscessus pulmonary disease and in only a little more than half of the infected patients, it is hard to see how this tool can be used clinically to detect M. abscessus earlier. One piece of data that might be helpful is to assess repeat pulmonary function testing in these patients, which might be more readily available than repeat chest CT scans, in order to see whether decline in lung function is correlated with increase in CT score.

Response

Thank you for encouraging and thoughtful comments. We will examine pulmonary function tests in a future study.

Specific Comment

Result section Lines 151 to 159. There is no need to repeat all the diagnoses in Table 1. Rather just say that 88% of the cohort had underlying pulmonary disease and leave it to the reader to see the distribution of disease in the table. Also for the lab values, it would be important to include the normal range.

Response

We deleted the repeated data (Line150-153), and added the normal range of the lab values in Table1.

The Clinical backgrounds of all Mab-PD patients were shown (n=32, Table1). Twentyeight patients (88%) had coexisting pulmonary diseases such as bronchiectasis, prior pulmonary tuberculosis, obstructive pulmonary disease (COPD, Asthma, bronchiolitis obliterans) and interstitial lung disease.

Comment 

Gaffky score. Please provide a reference for the Gaffky score. Also, in Table 1, I thin the line saying “Maximum number of Gaffky scale” should read the “Number of patients that had the maximum Gaffky scale of 10” and it is unclear what “total AFB culture positive count for Mab” means.

Response

We added a reference for the Gaffky score (reference number 13). We replace “Maximum number of Gaffky scale”to “Maximum Gaffky scale”. We also replaced “total AFB culture positive count for Mab” to “Number of times of AFB culture positive ”according to the reviewer’s comment in Table1.

Comment 

Table 2. The upper part of Table 2 repeats much of what is in Table 1 and needs to be deleted. In other words, begin Table 2 with “Time from pre-CT to diagnosis CT (days)” and then have the comparison between the two CT scan scores.

Response

We retained only necessary descriptions such as underlying diseases and deleted duplicated parts of Table 2 according to the reviewer’s comment.

---

## [Decision Letter · Decision Letter 1]

16 Jan 2023

The usefulness of change in CT score for evaluating the activity of Mycobacterium abscessus (Mab) pulmonary disease (Mab-PD)

PONE-D-22-29888R1

Dear Dr. Hara,

We’re pleased to inform you that your manuscript has been judged scientifically suitable for publication and will be formally accepted for publication once it meets all outstanding technical requirements.

Kind regards,

Frederick Quinn

Academic Editor

PLOS ONE

Additional Editor Comments (optional):

Reviewers' comments:

Reviewer's Responses to Questions

**Comments to the Author**

1. If the authors have adequately addressed your comments raised in a previous round of review and you feel that this manuscript is now acceptable for publication, you may indicate that here to bypass the “Comments to the Author” section, enter your conflict of interest statement in the “Confidential to Editor” section, and submit your "Accept" recommendation.

Reviewer #1: All comments have been addressed

2. Is the manuscript technically sound, and do the data support the conclusions?

Reviewer #1: Yes

3. Has the statistical analysis been performed appropriately and rigorously? 

Reviewer #1: Yes

4. Have the authors made all data underlying the findings in their manuscript fully available?

Reviewer #1: Yes

5. Is the manuscript presented in an intelligible fashion and written in standard English?

Reviewer #1: Yes

6. Review Comments to the Author

Reviewer #1: The authors have addressed my main concern and included additional information on a potential for an "outlier effect" in the revised discussion.

7. PLOS authors have the option to publish the peer review history of their article (what does this mean?). If published, this will include your full peer review and any attached files.

Reviewer #1: No

---

## [Editor Report · Acceptance letter]

23 Jan 2023

PONE-D-22-29888R1 

The usefulness of change in CT score for evaluating the activity of *Mycobacterium abscessus* (Mab) pulmonary disease (Mab-PD) 

Dear Dr. Hara:

I'm pleased to inform you that your manuscript has been deemed suitable for publication in PLOS ONE. Congratulations! Your manuscript is now with our production department. 

Kind regards, 

on behalf of

Dr. Frederick Quinn 

Academic Editor

PLOS ONE